# The Role of IL-33/ST2 in COPD and Its Future as an Antibody Therapy

**DOI:** 10.3390/ijms24108702

**Published:** 2023-05-12

**Authors:** Lluc Riera-Martínez, Laura Cànaves-Gómez, Amanda Iglesias, Aina Martin-Medina, Borja G. Cosío

**Affiliations:** 1Instituto de Investigación Sanitaria Illes Balears (IdISBa), Hospital Universitario Son Espases, 07120 Palma de Mallorca, Spain; 2Centro de Investigación Biomédica en Red de Enfermedades Respiratorias (CIBERES), Instituto de Salud Carlos III, 28029 Madrid, Spain; 3Department of Respiratory Medicine, Hospital Universitario Son Espases, 07120 Palma de Mallorca, Spain

**Keywords:** IL-33, COPD, clinical trials

## Abstract

COPD is a leading cause of mortality and morbidity worldwide and is associated with a high socioeconomic burden. Current treatment includes the use of inhaled corticosteroids and bronchodilators, which can help to improve symptoms and reduce exacerbations; however, there is no solution for restoring lung function and the emphysema caused by loss of the alveolar tissue. Moreover, exacerbations accelerate progression and challenge even more the management of COPD. Mechanisms of inflammation in COPD have been investigated over the past years, thus opening new avenues to develop novel targeted-directed therapies. Special attention has been paid to IL-33 and its receptor ST2, as they have been found to mediate immune responses and alveolar damage, and their expression is upregulated in COPD patients, which correlates with disease progression. Here we summarize the current knowledge on the IL-33/ST2 pathway and its involvement in COPD, with a special focus on developed antibodies and the ongoing clinical trials using anti-IL-33 and anti-ST2 strategies in COPD patients.

## 1. Introduction

Chronic obstructive pulmonary disease (COPD) is characterized by an abnormal inflammatory response of the lungs that causes emphysema in the lung parenchyma along with bronchitis in the upper airways, which translates into airflow limitation, impairing the breathing of the patients [1]. COPD affects approximately 384 million people worldwide, and it is expected to become the third leading cause of morbidity and mortality in the world population by 2030 [2,3]. In general, it is believed that COPD is due to pathophysiological changes caused by inhaling air pollutants, mainly cigarette smoke, along with intrinsic factors such as aging and genetic risk. The progression of the disease is often challenged by the sudden acuteness of the symptoms, the so-called exacerbations, which are mainly triggered by viral, bacterial pathogens such as *Haemophilus influenza*, *Moraxella catarrhalis*, and *Streptococcus pneumoniae* [4,5,6]. During COPD, as well as during COPD exacerbations, pro-inflammatory mediators such as cytokines and chemokines are activated, causing a sustained, harmful immune response [7]. Multiple studies have indicated that persistent inflammation can produce gradual changes in the airway structures that lead to obstruction and loss of lung epithelium leading to respiratory symptoms [8,9]. Among the pro-inflammatory cytokines, interleukin-33 (IL-33) is a cytokine of the interleukin-1 family that plays a key role in airway inflammation. In fact, it has recently been found to have a crucial role in patients with chronic obstructive pulmonary disease (COPD) [10]. For a long time, the treatment of COPD has been based on inhaled bronchodilators and corticosteroids; however, the discovery of new molecular mechanisms involved in the disease has allowed the recent development of more targeted drugs, including cytokine inhibitors [11]. Several reviews have discussed the biological and intracellular expression of IL33 and its extracellular regulation mechanism in detail. Therefore, in this review, we will briefly summarize the current knowledge of IL-33 and focus on its implication in COPD, highlighting its potential use as a target for antibody-mediated therapy.

## 2. IL-33 and Inflammation

The interleukin-1 (IL-1) superfamily of cytokines plays an important role in both the development of innate and adaptive immunity, thus regulating host defense, inflammation, and the management of injuries [12,13]. The IL-1 superfamily consists of 11 members: IL-1α, IL-1β, IL-1 receptor antagonist (IL-1Ra), IL-18, IL-33, IL-36α, IL-36β, IL-36γ, IL-36Ra, IL-37 and IL-38 [14,15]. All of the IL-1 family cytokines that are biologically active are extracellular molecules, while their precursors are mainly intracellular. The interleukin-1 (IL-1) family of cytokines and receptors share similar functions with the Toll-like receptor (TLR) family, another key player in the immune response [16], due to the high homology between IL-1 receptor Toll−interleukin-1 receptor (TIR) domain and TLR [16]. In the case of IL-33, the molecule called “Full-length human IL-33 protein” (IL-33FL) is a protein of 270 amino acids [10]. The IL-33FL protein comprises two evolutionarily conserved domains, the N-terminal nuclear domain, and the C-terminal IL-1-like cytokine domain, separated by a divergent “protease sensor” domain [17,18]. As mentioned above, IL33FL is a bioactive form of the protein; however, this form can be cleaved by cellular proteases, especially during inflammatory processes, resulting in shorter mature forms where the C-terminal domain is located. It has been shown that six different inflammatory serine proteases can cleave to IL-33FL, giving rise to shorter mature forms of 18–21 kDa. Serine proteases include neutrophil cathepsin G, elastase, and proteinase 3 (PR3), as well as mast cell chymase, tryptase, and granzyme B [19]. The mature forms of IL-33 generated by inflammatory proteases have significantly higher biological activity than their IL-33FL precursor [17]. In this regard, concentrations 10-fold higher are needed to obtain similar levels of IL-6 secretion by MC/9 mast cells and 30-fold higher to obtain similar levels of IL-5 or IL-13 by ILC2 [17,20]. The processing of IL-33FL has an important role in vivo, especially when IL-33FL is present in low concentrations or has little activity [20].

Regarding its cellular origin, IL-33 is predominantly found and produced by epithelial cells, fibroblasts, and endothelial cells [21]; however, its expression can also be induced in immune cells under certain inflammatory conditions [22]. Regarding biological function, IL-33 plays a dual-function role. On the one hand, it may have a role in tissue homeostasis while being sequestered in the nucleus, where it represses gene expression of certain inflammatory genes, and on the other hand, it is also involved in tissue repair and remodeling [23,24]. During inflammation and cellular necrosis, IL-33, which has been stored within the nucleus, is passively released, therefore playing an alarmin role, which is why IL-33 is included among the group of alarmins. In this scenario, IL-33 is processed by proteases and passes into its mature superactive form [25]. In contrast, during cellular apoptosis, IL-33 is retained within the cytoplasm of the cells and becomes inactivated by apoptotic proteases, such as caspase-3 and caspase-7 [10,26] (Figure 1 and Figure 2).

By playing its role as an alarmin, IL-33 alerts the immune system of damage [10] by binding to ST2, which is a member of the Toll-like receptor (TLR)/IL1R superfamily. The ST2 receptor is the main target of IL-33, and it is constitutively expressed by immune cells such as ILC2 [27], mast cells, and Tregs [28]. Moreover, IL-33 can also activate Th2 cells [29], basophils [30], eosinophils [31], M2 macrophages [32], and dendritic cells, which are involved in type 2 immunity and allergic inflammation [19,33,34]. Other immune cells that constitutively express ST2 and whose expression may be increased upon stimulation are natural killers (NK), invariant natural killer T cells (iNKT), and neutrophils [19]. Notably, ST2 is also expressed on goblet cells and epithelial cells, where they can further express IL-8 and IL-17F in an autocrine manner, leading to the amplification of inflammation [35,36]. Upon binding, the IL-33/ST2L complex binds to the ubiquitously expressed IL-1R accessory protein (IL-1RAcP), forming a heterodimer [37,38,39], and signaling is induced by dimerization of the Toll−interleukin-1 (TIR) cytoplasmic domains of IL-1RAcP and ST2. This leads to the recruitment of the adapter protein MyD88 and the activation of transcription factors such as NF-kappaB through TRAF6, IRAK-1/4, MAP kinases, and AP-1, producing the transcription of inflammatory mediators (Figure 3) [40].

## 3. IL33 and the Lung

In the lung, IL-33 expression has been observed mostly in airway epithelial basal cells, endothelial cells, and fibroblasts. Among these, IL-33 is predominately secreted by epithelial cells, which in turn can trigger the release of IL-33 by receptor cells, specifically by alveolar type 2 epithelial cells, endothelial cells, fibroblasts, and mast cells [41]. Secretion of IL-33 can be induced by inhaled stimuli, such as pathogens, allergens, cigarette smoke, and pollution, therefore triggering the activation of the immune response in the lung [42].

Some studies, including animal models and clinical data, have shown a critical role of IL-33 in allergic inflammation by driving persistent type 2 immune responses [34]. In chronic lung inflammation diseases, such as COPD or asthma, it has been suggested that increased IL-33 might have a role in maintaining the persistent inflammatory response by sustained upregulation of CD4+ T cells and ILC2 upon allergen exposure [43] and further upregulation of IL-33 in the lung epithelium [44]. In this context, IL-33 has been found to produce airway inflammation, hyperresponsiveness, and goblet cell metaplasia in allergen-naïve mice, and in allergen-exposed mice, it aggravates asthma-like responses [45]. Moreover, neutrophil and eosinophil levels, related to epithelial expression of IL-33 are increased in COPD and correlated with the severity stage [46]. In addition, it has been reported that IL-33 accelerates the maturation of hematopoietic progenitor cells (HPCs). These cells produce a large number of proinflammatory cytokines before differentiating into mature eosinophils at the sites of inflammation. For this reason, HPCs are considered effector cells in allergic asthma [47]. 

Under pathological conditions, IL-33 also seems to play an important role in airway remodeling. In this regard, stimulation of lung fibroblasts with recombinant IL-33 induces the secretion of collagen, and mice treated with anti-IL-33 antibodies are resistant to bleomycin-induced fibrosis [48]. The IL-33/ST2 pathway is also important in non-allergic inflammation as the resolution of inflammation for returning homeostasis for the function of tissue repair [49]. For instance, IL-33 can mediate lung tissue repair by inducing the activation of Tregs and ILC2s after influenza infection [50,51] (Figure 2). Li Q et al. also found that IL-33 causes the development of auto-antibodies against alveolar epithelial cells in mice and humans, implying that IL-33 can trigger an auto-immune response against lung tissue [41]. Preclinical models indicate that IL-33 is involved in airway remodeling, which happens as a result of persistent airway inflammation, such as COPD, and leads to irreversible loss of lung function [52]. Moreover, IL-33 has been involved in cigarette smoke (CS) induced lung damage. In this line, preclinical evidence suggests that CS exposure increases IL-33 production by airway epithelial cells, which is linked to higher vulnerability to virus-induced exacerbations in mice [53]. Another study found that CS causes an increase in the expression of the *IL-33* gene in lung epithelial cells and that viral infection increases the synthesis of IL-33 through an alternative IL-33-dependent inflammatory response in the airways, which is associated with type 1 immunological features and may play a role in COPD exacerbations [52,54].

## 4. IL-33 in COPD

Chronic obstructive pulmonary disease (COPD) is one of the most common respiratory diseases. COPD is characterized by an abnormal inflammatory response of the lungs [1]. Increased expression of IL-33 and ST2 receptors has been observed in COPD [55]; however, unlike in allergic diseases where it induces classic type 2 responses, the role and mechanisms are more complex. Cigarette smoke (CS) is a major driver of COPD development, and exposure to cigarette smoke can induce a chain of systemic responses that not only increase IL-33 production in epithelial and endothelial cells but also cause increased IL-33 expression in peripheral blood mononuclear cells (PBMC), which induces persistent activation of the immune system favoring COPD progression [56]. CS also alters ST2 receptor expression, which implies an increased type 1 pro-inflammatory response in the lungs, intensifying exacerbation-induced inflammation in COPD. It has been reported that IL-33 increases mucus production and vascular endothelial permeability, which further aggravates inflammation [57,58]. Lung infections commonly happen in the course of COPD, and using inhaled corticosteroids as a long-term therapy seems to increase their risk [59]. Infections might increase IL-33 expression and secretion in the epithelium and enhance IL-33 in PBMC and peripheral blood lymphocytes (PBL) [56] (Figure 4).

COPD is also characterized by a remarkable remodeling of the lung architecture consisting of thickening of the airways and distortion of the normal shape of the cells, including epithelial metaplasia, globet cell hypertrophy, and smooth muscle hyperplasia. Depending on the site where structural changes are happening, remodeling processes would contribute to emphysema of the lung parenchyma or to airway obstruction [60]. The exact mechanisms that drive structural changes in COPD are not fully understood; however, there is some evidence that immunomodulatory cells play an important role [61]. In fact, IL-33, along with other alarmins, have an important role in tissue damage and remodeling in this condition [10,56,62]. Both Th1 inflammatory cytokines such as TNF-alpha, IL-1beta, IL-6, IL-8, and IL-12 [63], as well as Th2 cytokines such as IL-5 [64], have been found to increase in COPD and to correlate with an increased risk of exacerbations. Interestingly, there is an increased expression of IL-6 and IL-8 stimulated by IL-33 in lung epithelial and endothelial cells via ST2/Il-1RacP and MAPKS pathway [56,62,65].

In humans, high IL-33 and its receptor ST2 expression are found in whole lung homogenates, as well as in epithelial and endothelial cells in lung biopsies from patients with COPD [55,57]; moreover, it has been detected in PBMC in COPD patients compared to healthy donors [66]. Lung IL-33 expression is also increased in animal models of cigarette smoke-induced COPD, where it has been found to have a role in the production of systemic inflammation [65].

However, the precise role of the IL-33/ST2 pathway in the pathogenesis of COPD is unclear. Recent studies suggested that IL-33 and its receptor ST2 may be considered critical factors during this condition mediating tissue remodeling in COPD by inducing lung collagen deposition [67]. In the study from Joo et al., IL-33 levels were measured in 62 COPD individuals and prospectively followed up for 1 year; moreover, they measured IL-33 expression in lung tissue from 38 patients. They found a positive correlation between levels of IL-33 in plasma and risk of exacerbation, and higher IL-33 expression in lung tissue correlated with worse lung function [68]. Similar to them, others also found higher IL-33 in serum [55], breath condensate [47], and lung tissue [53] from COPD patients in a stable stage compared to controls, the latter also inversely correlated with lung function. Conversely, there is one study that reported that during exacerbations, IL-33 concentrations in serum are lower, which, they suggest, may represent the decreased activity of Th2 cells during these acute events [63]. Bhowmik et al. and Bucchioni et al. also reported an increased expression in bronchoalveolar lavage (BAL) and lungs of IL-6 and IL-8 in COPD patients, mediating recruitment of neutrophils, T-cells, and macrophages to the lungs which leads to lung tissue damage, and contributes to the elastase and protease imbalance that promotes loss of epithelium causing lung emphysema [62,69]. Moreover, IL-33 has also been found to mediate alveolar epithelial cell apoptosis, contributing to emphysematous damage [70,71,72,73]. Apart from that, eosinophil recruitment has been recently involved in COPD by contributing to the inflammatory response [69], and IL-33 levels in exhaled breath condensate correlated with blood eosinophil count in COPD [47]. Therefore, targeting IL-33 through different inflammatory pathways may be an advantage in treating some lung diseases such as asthma and COPD [10,74].

Apart from those caused by bacteria, respiratory viral infections are now considered another major cause of acute worsening of symptoms due to COPD exacerbations. Walzl, G. et al. were the first to describe the role of IL-33 during Respiratory Syncytial Virus infection (RSV) by exposing mice to the virus plus treatment with the ST2 antibody, which decreased eosinophilic inflammation, Th2 cytokine levels in BAL, and induced viral clearance [75]. Another preclinical study found that infection of mice with influenza activates the IL-33/IL-13 axis, thus inducing airway inflammation mainly upon activation of macrophages and neutrophils [76]. Given that current treatments for viral infections are based on preventive measures such as yearly vaccination, targeting IL-33 might represent a novel strategy to halt or reduce the dramatic effects of COPD exacerbations due to these pathogens.

## 5. Targeting IL-33 in COPD

Despite the fact that much effort has been made in the improvement of COPD management in clinics, current treatments are still based on inhaled corticosteroids and bronchodilators. This kind of medication is beneficial in reducing the acuteness of the symptoms; however, it does not improve lung function or stop disease progression. In the last 10 years, there has been an increasing interest in the use of biologics against cytokines and/or their receptors in the field of respiratory diseases, such as asthma and COPD. In agreement, antibodies against the most relevant cytokines have been developed, for instance, anti-IL-4/13 receptor (dupilumab), anti-IL-5 (mepolizumab, reslizumab) or anti-IL-5 receptor (benralizumab), and anti-IL-13 (lebrikizumab and tralokinumab) and anti-thymic stromal lymphopoietin (TSLP; tezepelumab), which have demonstrated efficacy in asthma patients who have been well stratified and whose endotypes have been correctly identified [77,78,79]. In COPD, however, despite being promising, anti-cytokine therapy is still under development. 

Anti-IL-33 has been shown to have beneficial effects in preclinical models of lung inflammation. In this regard, lung damage was reduced after treatment with soluble ST2 (sST2) in LPS- or bleomycin-treated mice [52,54,70]. In another experimental mouse model using house dust to induce lung inflammation in which ST2 gene expression is increased, inhibition of IL-33 by itepekimab treatment reduced inflammation of the airways and further restored ST2 levels [80,81]. This data supports blocking the IL-33/ST2 pathway as a therapeutic approach for inflammatory lung diseases where a role for IL-33/ST2 has been reported.

In COPD patients, increased levels of IL33 in serum correlate with a higher risk of exacerbations, and higher IL-33 expression in lung tissue correlates with disease severity [53,68] and triggers sustained immune responses causing alveolar epithelial cell damage, which makes the idea of targeting IL-33 by using antibody-mediated blockades [77,78,79] attractive.

In this regard, clinical trials using anti-IL-33 and anti-ST2 monoclonal antibodies have begun to be evaluated for tolerability and effectiveness in COPD treatment (Table 1).

Currently, there are 12 registered clinical trials directed against IL-33 and its receptor ST2. The first was the proof of concept study NCT03546907, which showed that SAR440340 (anti-IL-33 mAb) is safe and well tolerated in patients with moderate-to-severe COPD. The hypotheses of this study were that genetic variants of the IL-33 pathway were also associated with COPD. The inclusion criteria of this clinical trial were current or former smokers between 40 and 75 years of age diagnosed with COPD for at least 1 year who were on a stable regimen of double or triple inhalation-based maintenance therapy. They were randomly assigned to receive either itepekimab 300 mg or a placebo administered as two subcutaneous injections every 2 weeks for 24–52 weeks. The main objective of this trial was to evaluate the annual rate of moderate and severe exacerbations in COPD patients during the treatment period. In the analysis of the results, they reported that genetic analysis showed an association between the loss of IL-33 function and a lower risk of COPD progression. Unfortunately, the primary endpoint in the total population was not reached, but when performing a subgroup analysis, they found that SAR440340 reduced the rate of exacerbations and improved lung function in former smokers with COPD [82].

The ongoing NCT05326412 clinical trial is a two-part, 12-week, exploratory phase 2a study. In this trial, the investigators aim to evaluate the mechanism of action of itepekimab (anti-IL-33 mAb) and its impact on airway inflammation in former and current smokers with COPD diagnosed at least one year and aged between 40 and 70 years. Participants have to be under standard treatment with monotherapy (long-acting β2-agonist [LABA]) or long-acting muscarinic antagonist [LAMA]), dual therapy (inhaled corticosteroid [ICS] + LABA, LABA + LAMA or ICS + LAMA), or triple therapy (ICS + LABA + LAMA) for COPD for at least 3 months prior to the screening visit and stable for at least 1 month before and during the screening. In addition, participants will be able to continue taking their COPD control medication throughout the study period, with the exception of systemic corticosteroids and/or antibiotics used for acute exacerbation of COPD (AECOPD). In this trial, there is an estimated number of enrollments of 60 patients, which will be further divided into 3 populations: smokers (Part A), former smokers (Part B), and current smokers (Part B), and it is estimated to last a total of 36 weeks between Part A and B (4-week screening period plus 12-week treatment period. Itepekimab will be administrated subcutaneously every 2 weeks and the patients will undergo a 20-week follow-up period [83]. The same drug (itepekimab) is currently being tested by two more phase 3 clinical trials (NCT04701983 and NCT04751487) to test the efficacy, safety, and tolerability in former smokers with moderate-to-severe COPD [84].

In addition, astegolimab is an IgG2 anti-ST2 monoclonal antibody directed against the IL-33 receptor (ST2) used in asthmatic patients to reduce their exacerbations [81]. There are currently three ongoing clinical trials to test this mAb in COPD patients. The objective of the clinical trial NCT03615040 is to evaluate whether astegolimab will reduce exacerbations in patients with moderate to very severe COPD. Inclusion criteria were participants with a clinical diagnosis of COPD of at least 1 year, older than 40 years, with moderate to very severe airflow obstruction, and who had suffered at least 2 acute exacerbations of COPD (defined as those requiring oral corticosteroids, antibiotics, or hospital admission) in the previous year. Participants received 490 mg of subcutaneous astegolimab or subcutaneous placebo every 4 weeks for a total of 44 weeks. The results showed that using astegolimab in patients with moderate to severe COPD did not significantly reduce the rate of exacerbations but did improve health status compared with the placebo [82]. Two more clinical trials using astegolimab will test its efficacy in reducing the rate of frequent exacerbations. The NCT050505037929 trial studies the efficacy, safety, and pharmacokinetics of astegolimab in combination with standard treatment in patients with COPD who are current or former smokers and have a history of frequent exacerbations. Moreover, the NCT05595642 trial aims to evaluate the efficacy and safety of astegolimab compared to the placebo in participants with COPD who are or have been smokers and have a history of exacerbations. In both trials, 1 group of participants will receive a subcutaneous injection of astegolimab or a placebo every 2 weeks, whereas another group will be administered the treatment every 4 weeks [85,86].

The clinical trials using tozorakimab, a human therapeutic antibody with a dual pharmacological profile that inhibits IL-33 activities through the ST2 and RAGE/EGFR signaling pathways, found a decrease in excess inflammation and epithelial remodeling in diseases caused by IL-33 [87]. With the phase 1 trial (NCT03096795), they wanted to observe the safety, tolerability, pharmacokinetics, and immunogenicity of MEDI3506 (later named tozorakim) by administering it as a single ascending dose in healthy subjects with atopy by giving multiple ascending doses in subjects with COPD and as a single dose in healthy Japanese subjects [88]. In the NCT04631016 trial, the goal is to observe the efficacy and safety of the drug MEDI3506 for treating adult subjects with moderate to severe COPD and chronic bronchitis. [89]. The aim of this Phase 3 study (NCT05166889) is to evaluate the efficacy and safety of dose 1 and dose 2 of tozorakimab administered subcutaneously (SC) in adult participants with symptomatic COPD and a history of ≥2 moderate or ≥1 severe exacerbation of COPD in the previous 12 months. Participants must receive optimized treatment with maintenance inhaled therapy (triple therapy with ICS/LABA/LAMA or dual therapy if triple therapy is not deemed appropriate) at stable doses for at least 3 months prior to enrollment [90]. NCT05158387 is the same as above, with the difference that patients can take dual therapy if triple therapy is not considered appropriate [91]. NCT05742802 is an extension of the 2 previous studies to evaluate the efficacy and safety of tozorakimab versus the placebo in adult participants (over 40 years of age) with symptomatic COPD and a history of exacerbations [92].

Taken together, the clinical trials of astegolimab and itepekimab have generated evidence and enthusiasm for the antibody-mediated strategy, uncovering potential responder subgroups [82,93]. Hopefully, upcoming clinical trials that should also include patients with milder diseases to explore the role of IL-33 in COPD activity and progression will provide definitive answers.

## 6. Conclusions

Management of COPD progression is still challenging due to limited current therapies that are not capable of reversing lung damage and the exacerbations that most of the patients eventually experience. Recent interest has been directed towards the development of more targeted therapies against specific molecules taking part in those processes as this represents a new option for COPD treatment. The IL-33/ST2 pathway has been demonstrated to orchestrate, at least in part, the inflammatory and remodeling processes taking part in COPD. However, the mechanisms by which IL-33/ST2 is involved in the disease are more complex and heterogeneous than those already reported in allergic airway inflammation (like in asthma), meaning that its role in COPD is rather defined by the context and progression of the disease which definitely deserves more research. Recent phase 2 and 2a clinical trials using antibodies against IL-33 or ST2 have demonstrated that targeting anti-IL-33 is safe and can improve lung function and reduce the risk of exacerbations in moderate-to-severe and very severe COPD patients compared to the placebo. Ongoing phase 3 studies will demonstrate the efficacy of itepekimab in former smokers with moderate-to-severe COPD. In addition, astegolimab, an anti-ST2 antibody that has already proven efficacy in asthma, is being tested in current and former smoker COPD patients that have suffered from exacerbations in three clinical trials. The results from these trials will definitely be relevant to decide on the future implementation of anti-IL-33/ST2 therapy in clinics.

## Figures and Tables

**Figure 1 ijms-24-08702-f001:**
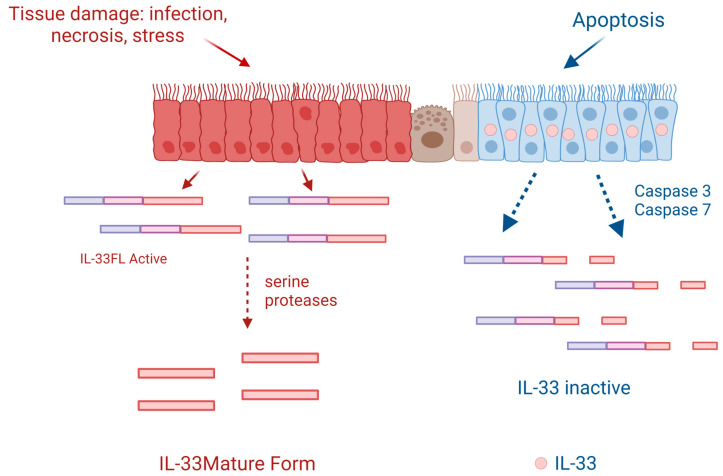
Following cell injury or necrosis, IL-33-producing cells release the full-length human IL-33 protein (IL-33FL) that can be cleaved extracellularly by proteases from inflammatory cells such as neutrophils or mast cells, passing to a mature form that has an activity of 10 to 30 times greater. When cells are programmed for apoptosis, IL-33 is retained in their nucleus and becomes inactivated if released into the cytoplasm by caspases 3 and 7. Created with BioRender.com.

**Figure 2 ijms-24-08702-f002:**
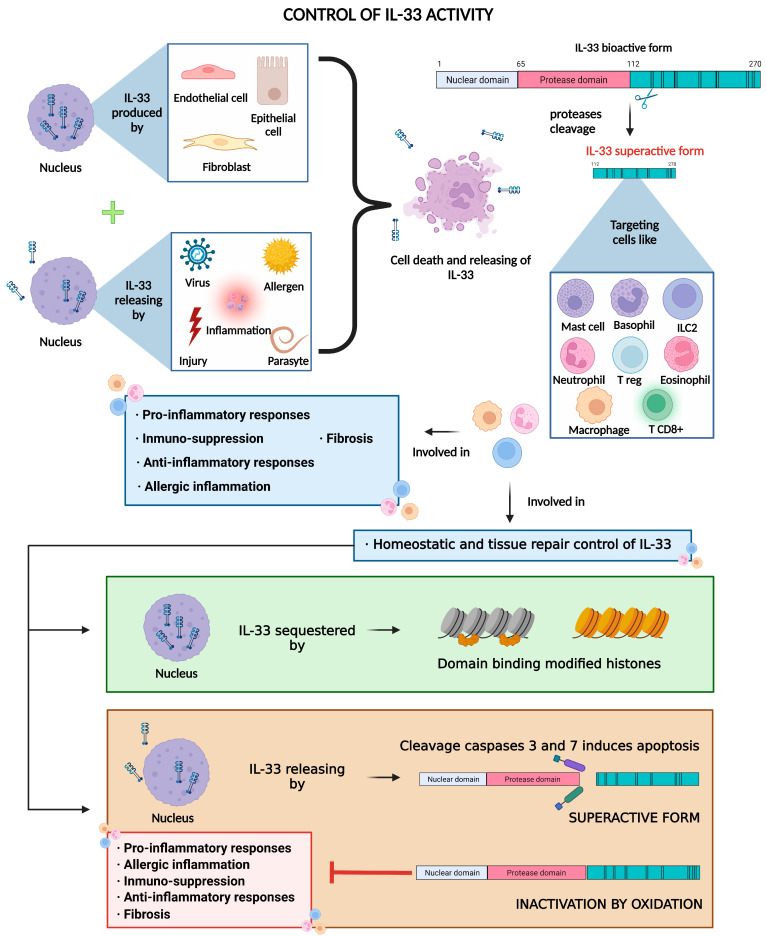
IL-33 is constitutively produced and stored in the nucleus of lung cells (mainly endothelial and epithelial cells and fibroblasts) and then released due to cell death in response to cell insults such as chronic inflammation or exposure to pathogens and allergens. Free full-length IL-33 is already bioactive, but it can be cleaved by proteases into more active forms, which in turn, bind to the ST2 receptor, mainly present in immune cells, thus mediating immune response, homeostatic control, and tissue repair processes. Created with BioRender.com.

**Figure 3 ijms-24-08702-f003:**
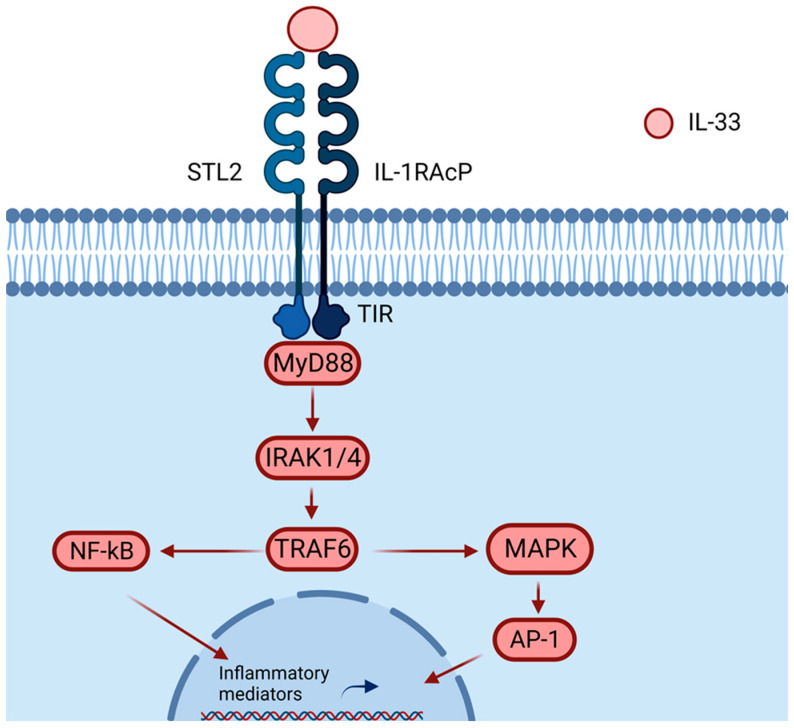
IL-33 binds to the heterodimer formed by ST2/IL-1RAcP, recruiting MyD88 into its intracellular domain (TIF). MyD88 binding recruits IL-1R-associated kinase (IRAK) and TRAF6, activating the NF-κB or AP-1 pathway. Activations of NF-κB and AP-1 promote inflammatory cytokine expressions. Created with BioRender.com.

**Figure 4 ijms-24-08702-f004:**
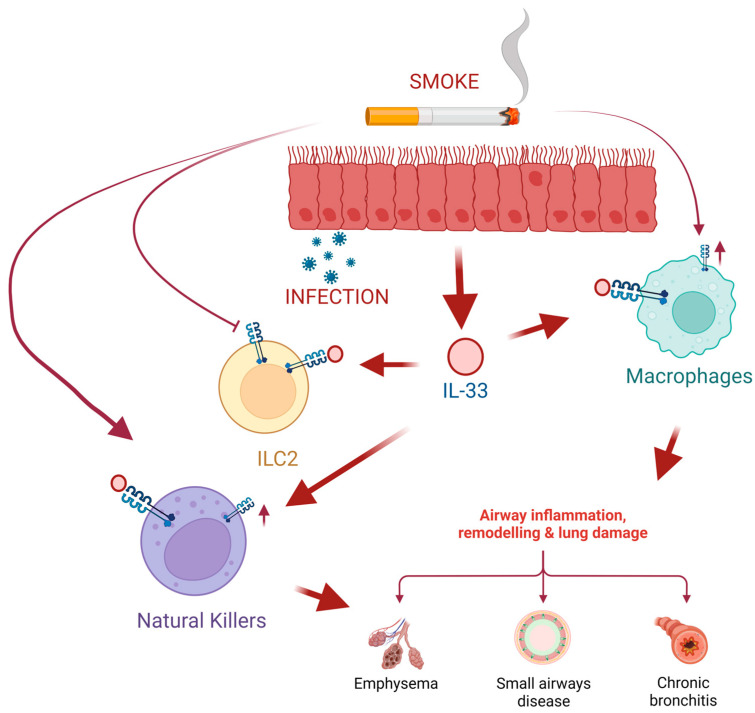
Cigarette smoke induces lung epithelial cells to produce high levels of IL-33 that will be released after cell damage, such as viral or bacterial infections. On the one hand, exposure to smoke inhibits the expression of the IL-33 (ST2) receptor in type 2 innate lymphoid cells (ILC2), which leads to the inhibition of ILC2 responses and the synthesis of cytokines induced by IL-33. Moreover, tobacco smoke favors the expression of ST2 in macrophages (M0) and natural killer (NK) cells. This leads to NK cell proliferation and increased production of the pro-inflammatory cytokine IFN-gamma leading to exacerbated COPD. Created with BioRender.com.

**Table 1 ijms-24-08702-t001:** Summary of ongoing and completed clinical trials with anti-IL-33 and ST2 monoclonal antibodies in COPD.

Molecule	Phase	Population	ClinicalTrials.gov
Anti-IL33 SAR440340	Phase 2	Moderate-to-severe acute exacerbations of Chronic Obstructive Pulmonary Disease	NCT03546907
Anti-IL33 Itepekimab	Phase 2a	Chronic Obstructive Pulmonary Disease	NCT05326412
Anti-IL33 Itepekimab/SAR440340	Phase 3	Chronic Obstructive Pulmonary Disease	NCT04701983
SAR440340/REGN3500/Itepekimab	Phase 3	Chronic Obstructive Pulmonary Disease	NCT04751487
Anti-ST2 Astegolimab	Phase 2a	Moderate-to-very severe Chronic Obstructive Pulmonary Disease	NCT03615040
Anti-ST2 Astegolimab	Phase 2b	Chronic Obstructive Pulmonary Disease	NCT05595642
Anti-ST2 Astegolimab	Phase 2	Chronic Obstructive Pulmonary Disease	NCT05037929
MEDI-3506/Tozorakimab	Phase 1	Chronic Obstructive Pulmonary Disease and Healthy Japanese Participants	NCT03096795
MEDI-3506/Tozorakimab	Phase 2	Chronic Obstructive Pulmonary Disease and Chronic Bronchitis.	NCT04631016
Tozorakimab	Phase 3	Chronic Obstructive Pulmonary Disease	NCT05166889
Tozorakimab	Phase 3	Chronic Obstructive Pulmonary Disease	NCT05158387
Tozorakimab	Phase 3	Chronic Obstructive Pulmonary Disease	NCT05742802

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
