# Peer review of "The Role of IL-33/ST2 in COPD and Its Future as an Antibody Therapy"

_ijms, 2023, doi:10.3390/ijms24108702_

Round 1

Reviewer 1 Report

The authors coherently summarized the behavioral patterns of IL-33/ST2 in inflammatory responses and in COPD evolution through introduction of current molecular biology and clinical studies. It is quite complicated to consult literature and clinical trials, however, there are a few concerns related to the content and use of language.

General comments

1.    Several reviews have discussed the biological and intracellular expression of IL33 and its extracellular regulation mechanism in detail. The most innovative part of the article is limited to the clinical research of anti-IL-33 antibodies.

2.    Several antibodies are introduced,  but the manuscript focuses on the controlled design of clinical trials, indicating the reliability of trials. The scope of description can be extended to preclinical studies and deficiencies of existing clinical research.

3.    There are many simple typos and grammatical inaccuracies between the lines, especially sections under the two headings ‘IL-33 in COPD’ and ‘targeting IL-33 in COPD’, which weaken the manuscript’s professionalism.

4.    The resolution of figures is expected to be improved。

5.    The citation style must be unified.

Comments

1.    A certain number of typos exist and repeat in the manuscript.

-“enhace” should be changed to “enhance ” (line 172).

-“archytechture” should be changed to “architecture” (line 182).

- “dysortion” should be changed to “distortion” (line 183)

-“osbtruction” should be changed to “obstruction” (line 186).

-“epihtelium” should be changed to “epithelium” (line 218).

-“fromt” should be changed to “from” (line 227).

-“clearence” should be changed to “clearance” (line 232).

-“influeza” should be changed to “influenza” (line 232).

-“benefitial” should be changed to “beneficial” (line 241).

-“deveolped” should be changed to “developed” (line 245).

-“exarcebations” should be changed to “exacerbations” (line 193,208,228,251-252,309,333-334,). There are also correct spellings of ‘exacerbations’ in the article, the format is not uniform.

-line182-195: ‘remodeling’ and ‘remodelling’ occur in the same paragraph.

2.       Figure 3 & 4: These two pictures can be combined into one. Based on Figure 3, we can include disease inducements, cell-cell interactions and physiological effects of immune response.

3.     The sequence number of the "Conclusions" section should be 6.

4.     Line 83-92: These sentences described three different conditions that IL-33 is involved, but Figure 1 only displays the second and third one.

5.     Line 121-124: There seems some grammatically wrong. And the information conveyed by this sentence is confused.

6.     Line 129: A spelling mistake, “COPR”?

7.     Line 143: It seem to be WT mice rather than ST2 deficient mice who received anti-IL-33 antibody.

8.     Figure 3: The responses of IL-33 stimulated cells can be divided into two sides, as in line 84 “IL-33 is a dual-function cytokine”.

9.     Figure 4: The arrow from nature killer cells to airway inflammation and other pathological changes is not that optimal, since other cells also participate in this process.

10.  Line 219-222: The sentences are repeated twice.

11.  Table 1: IL-33 antibodies are mainly test in moderate-to-severe COPD, is it useless to prevent the deterioration of mild COPD?

12.   Line 289-290: The groups of participants are not in accordance with the information of the trial(https://clinicaltrials.gov/ct2/show/NCT05326412).

Reviewer 2 Report

This is a very interesting and well written review about the current knowledge on IL-33/ST2 pathway and its involvement in COPD, with a special focus on developed antibodies and the ongoing clinical trials using anti-IL-33/ST2 strategies in COPD. The review is suitable for publication in IJMS but authors should improve the quality of all illustrations showed.

Reviewer 3 Report

Nicely written and almost comprehensive review. For COPD all the tozorakimab studies are missing in the table e.g. NCT05166889 and other, please also see Tozorakimab: a dual-pharmacology anti-IL-33 antibody that inhibits IL-33 signalling via ST2 and RAGE/EGFR to reduce inflammation and epithelial dysfunction. There are three studies  I C ScottE EnglandD G ReesT ErngrenC Chaillan HuntingtonK F HouslayD A SimsC HollinsE C HinchyC ColleyD J CorkillE S Cohen

Round 2

Reviewer 1 Report

Authors have answered all the comments from previous reviewers' reports. And this versionis much better, and is good for publihing in IJMS.